# Nonlinear Robust Control of Vehicle Stabilization System with Uncertainty Based on Neural Network

Yimin Wang [1], Shusen Yuan [2,*], Xiuye Wang [1] and Guolai Yang [1]

1  School of Mechanical Engineering, Nanjing University of Science & Technology, Nanjing 210094, China; wymin@njust.edu.cn (Y.W.); xiuyewang@njust.edu.cn (X.W.); yyanggl@njust.edu.cn (G.Y.)
2  National Key Laboratory of Transient Physics, Nanjing University of Science and Technology, Nanjing 210094, China
*  Correspondence: njustyuan@163.com

**Abstract:** To effectively suppress the effects of uncertainties including unmodeled dynamics and external disturbances in the vehicle stabilization system, a nonlinear robust control strategy based on a multilayer neural network is proposed in this paper. First, the mechanical and electrical coupling dynamics model of the vehicle stabilization system, considering model uncertainty and actuator dynamics, is refined. Second, the lumped uncertainty of the vehicle stabilization system is estimated by a multi-layer neural network and compensated by feedforward control. The high robustness of the system is ensured by constructing the sliding mode feedback control law. The proposed control method overcomes the limitations of sliding mode technology and the neural network and is naturally applied to the vehicle stabilization system, avoiding the adverse effects of high-gain feedback. Based on Lyapunov theory, it is demonstrated that the proposed controller is able to achieve the desired stability tracking performance. Finally, the effectiveness of the proposed control strategy is verified by co-simulation and comparative experiments.

**Keywords:** vehicle stabilization system; sliding mode control; neural network; robust control; lumped uncertainty

## 1. Introduction

The vehicle stabilization system studied in this paper is a core component for reducing tank gun barrel jitter and improving tank aiming accuracy [1]. It ensures the precise pointing motion of the tank gun in the vertical direction by tracking the target commands given by the fire control system in real time [2]. In the traditional tank vehicle stabilization system, the actuator of the tank gun is a hydraulic cylinder, which inevitably suffers from the defects of running, dripping, and leaking, and there is a torque transfer hysteresis, which makes it difficult to further improve the tracking performance and stabilization accuracy of the tank gun in a high-mobility environment [3,4]. Therefore, the new generation of tank vehicle stabilization systems fully replaces the traditional actuator hydraulic cylinder of the tank gun by an electric cylinder. In the process of analytical dynamics modeling of the tank vehicle stabilization system, the state-space equation has been completely changed, although the hydraulic transmission characteristics are not considered [5]. Under the high-mobility driving environment of tanks, the coupling, nonlinearity and uncertainty of the vehicle stabilization system are more prominent, and it is difficult to establish high-accuracy analytical dynamics equations and design high-performance control strategies [6,7].

In order to further improve the control performance of the vehicle stabilization system under the high-speed driving environment of tanks and ensure the motion-to-motion striking accuracy of the tank gun, researchers have been focusing on the fine dynamic modeling and nonlinear robust controller design of the system. The mechatronic dynamics model of the tank vehicle stabilization system is developed, but the effects of unmodeled

disturbances and other uncertainties of the system are neglected [8]. The coupling, nonlinearity, and uncertainty of the tank vehicle stabilization system are considered in [9], but the essence of its dynamic modeling is only its mechanical dynamic modeling, without considering the electrical dynamic modeling of the actuator, which does not conform to the actual driving mechanism of the vehicle stabilization system. A comprehensive mathematical dynamics equation for the tank vehicle stabilization system is effectively demonstrated in [10]. However, it neglects the mechanism's nonlinearity in the linear actuator driving process. Specifically, the angular transition relationship between the linear motion of the linear actuator and the rotational motion of the tank gun is unclear. For this reason, this paper first expects to lay the model's foundation for subsequent research of control methods by accurately establishing an integrated dynamics model that considers the electromechanical coupling dynamics of the vehicle's stabilization system, the nonlinearities in the mechanism of the linear actuator and the aggregate uncertainty of the system.

In terms of the expansion of control methods, the traditional research on tank vehicle stabilization control technology is mainly limited to the firing accuracy of the tank gun between low-speed travel and a stationary state. Therefore, when designing the controller, the controlled system is generally simplified as a linear system, and PID [11,12] or LQR control [13,14] is selected. However, in the new generation of tanks, the requirement of firing while traveling at high speed has been proposed to improve combat capabilities. Since the uncertainty of the system is time-varying (and may be fast), more suitable controllers need to be designed for such strongly nonlinear control systems. In the literature [15], the design process of two robust H∞ control architectures for a gun-launched micro aerial vehicle is presented. The process starts with the development of a nonlinear dynamic model reflecting the important elements. This model is then linearized around hover and used for the design of static output feedback H∞ controllers for position and orientation control. However, there is a significant amount of nonlinearity in the vehicle stabilization system. The use of linear controllers that are not model-based will result in an increased computational burden and increased delay. It further causes the signal response of the control system to lag and does not meet the control requirements of the system well. With the continuous development of modern control theory, various nonlinear robust control methods are gradually applied to tank vehicle stabilization systems [16]. Chen et al. developed an adaptive controller to achieve asymptotic tracking for tank vehicle stabilization systems [17]. Unfortunately, if there is unmodeled uncertainty in the vehicle stabilization system, the adaptive control is prone to destabilization and cannot guarantee the stability of the system. In the literature [18,19], the combination of adaptive control and deterministic robust control is perfectly applied to the tank vehicle stabilization system, which ensures that the system theoretically achieves bounded stability even if there are various uncertainties and improves the robustness of the controller. Inspired by adaptive robust control, Ma et al. proposed an adaptive robust control based on the constraint-following method for tracking a moving tank target [20,21]. The controller adopts the concept of constraint-following, which uses the time-varying pointing angle as a constraint to adjust the barrel from the initial position to the target position. In addition, the adaptive rate can compensate the error term in the control process and improve the stability of the system around the ideal tracking angle. To further deal with the uncertainties in tank vehicle stabilization systems, extended state observers [22,23] and time-delay estimation techniques [24] have also been applied to the precise pointing motion of tank guns. In [22,23], the design of the disturbance observer is used to estimate and compensate the mechanical dynamic uncertainty of the vehicle stabilization system, which is an effective way to reduce the influence of friction nonlinearity and structural flexibility of the mechanical system to a certain extent. According to [24], the complex modeling of the mechanical dynamics of the system is avoided by the time-delay estimation technique, the system's lumped uncertainty is estimated, and the finite-time stabilization of the tank vehicle stabilization system is obtained by incorporating sliding mode control. The above control methods promote further improvement in control performance of vehicle stabilization systems and

are effective treatments for uncertainty in vehicle stabilization systems. However, they all treat the uncertainty of the system by lumped disturbance and inevitably have the influence of high-gain feedback; it is easy to stimulate the unmodeled high-frequency dynamics of the tank vehicle stabilization system in high-mobility environments, affecting the stability accuracy of the tank gun.

Due to the complexity of tank vehicle stabilization systems, it is necessary to design a high-performance control strategy, which can not only effectively estimate the unmodeled dynamics and various uncertainties of the system, but also ensure high robustness in environments requiring complex maneuvering. In recent years, with the development of learning control and numerical technology, the concept of intelligent learning control has received wide attention, forming the apex of neural network (NN) development. NNs are highly robust and fault-tolerant [25] and are capable of processing data in parallel. Therefore, the calculation time is greatly reduced, which is suitable for the uncertainty suppression of tank vehicle stabilization systems. In [26], based on the approximation performance of a neural network, adaptive control strategy and a neural network are combined to compensate the uncertainty of a tank vehicle stabilization system. Although this method can obtain an ideal asymptotic tracking effect, it does not consider its electrical dynamics and only estimates the mechanical uncertainty of the system. Meanwhile, the mathematical simulation presented in [26] cannot model road excitation, and it is difficult to truly reproduce the impact of strong impact interference during vehicle movement. Another advantage of neural networks is the fact that they can be used as black box estimators [27]. Its self-learning, self-organization, and self-adaptability allow it to handle unknown models, such as those exhibiting nonlinear relationships in complex systems [28]. The excellent approximation performance of neural networks in system nonlinearities and uncertainties has led to their wide application, facilitating improvements in the tracking performance of systems such as automobile engines and hydraulic actuators [29]. With the more stringent requirements of modern warfare on the high-precision shooting of tank guns, intelligent control algorithms represented by neural networks are gradually being applied to the tank vehicle stabilization system. Aiming to resolve issues in trajectory tracking in the tank vehicle stabilization systems, reference [30] proposes an iterative learning control scheme of error tracking based on neural networks, and numerical experiments demonstrate its effectiveness. This further inspires this paper to combine neural networks with nonlinear robust control and make use of the approximation ability of neural networks owing to its arbitrary nonlinear characteristics, in order to explore the high-performance control of tank vehicle stabilization systems exhibiting coupling, nonlinearity, and uncertainty. Therefore, this paper applies the neural network method and the sliding mode control strategy to the tank vehicle stabilization system. The uncertainty and nonlinearity of the system are estimated by multi-layer neural network and compensated by feedforward control. The sliding mode feedback control law is used to ensure the robustness of the system. Under the guidance of this control concept, the tracking accuracy of the vehicle stabilization system is expected to be further improved.

Inspired by the above, this paper first establishes a high-precision electromechanical coupling dynamic model with uncertainty based on the potential energy and kinetic energy of the vehicle stabilization system. The modeling process analyzes the nonlinearity in the mechanism of the linear actuator that converts linear motion into rotary motion in the tank gun. Second, the intelligent learning control method is introduced into the vehicle stabilization system. A three-layer neural network is used to approximate the unmodeled disturbance and unknown dynamics of the system. The sliding mode feedback control law is constructed to ensure the robustness of the controller design and enhance engineering applicability. On this basis, through the stability analysis of the Lyapunov function, it is proved that the proposed sliding mode robust controller based on neural network compensation can obtain favorable stable tracking performance even with various uncertainties. Finally, the multi-body dynamics model and the semi-physical simulation model of the vehicle stabilization system are established to fully simulate various uncertainties in the

high-mobility environments faced by tanks. The effectiveness of the proposed control method is verified by a large number of co-simulation and comparative experiments.

## 2. System Description and Model Establishment

The tank vehicle stabilization system is key in ensuring accurate tracking and stable aim of the tank gun in high-speed-driving environments. Improving the pointing accuracy of tank guns is the core problem discussed in this paper. Tanks have special characteristics that are different from those of general mechanical systems. They have very strict requirements on their controllers, which need to have extremely high robustness and extremely fast tracking accuracy. It is necessary not only to perfect the control theory, but also to analyze the dynamic transfer process in detail. In order to ensure the effectiveness of model-based control strategy design, this paper first analyzes the dynamic characteristics of the vehicle stabilization system under strong disturbance and establishes the electromechanical coupling dynamic model of tank vehicle stabilization systems with uncertainty.

**Remark 1.** *In the vehicle stabilization systems of real-world tanks, uncertainties including parameter uncertainties and uncertain nonlinearities embodied in the vertical direction are far more complex than those in the horizontal direction, including gaps, couplings, etc. This paper is based on the study of the tank vehicle stabilization system in the pitch direction, with its horizontal direction kept locked. A mechatronic modeling and controller design study for the pitch direction of the tank vehicle stabilization system is conducted, while the horizontal direction is kept in the locked state. These works are sufficient to contribute to the improvement in the pointing accuracy of real-world tanks and have important research and engineering value.*

The tank vehicle stabilization system considered in this paper is presented in Figure 1, which clearly provides the structural components and geometric arrangement of the system. For the convenience of the potential energy description, we define the zero-degree pointing angle as the zero-potential energy position of the tank vehicle stabilization system, and notably, this is also the initial position. The symbols involved in Figure 1 are explained as follows. The length from the rotation center of the trunnion to the upper pivot point of the electrical cylinder is $L_1$, and the length to the lower pivot point of the electrical cylinder is $L_2$. The initial length of the electrical cylinder is $L_0$. The initial angle of the electrical cylinder is the acute angle $\alpha$ between $L_1$ and $L_2$. $q$ is the angular displacement of the tank gun barrel. $\Delta L$ is the extension length of the electrical cylinder.

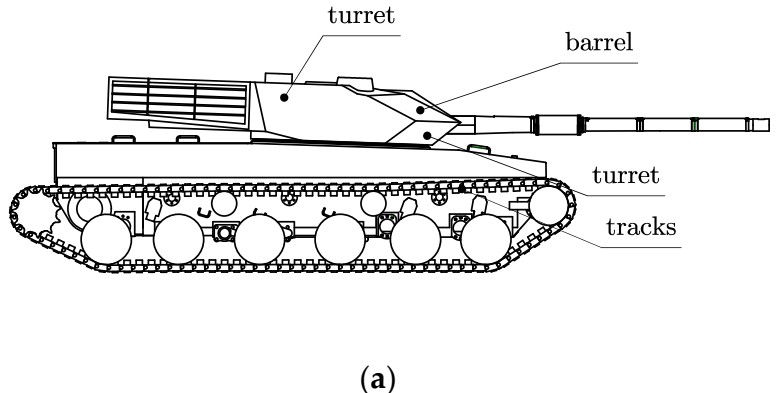

(**a**)

**Figure 1.** *Cont.*

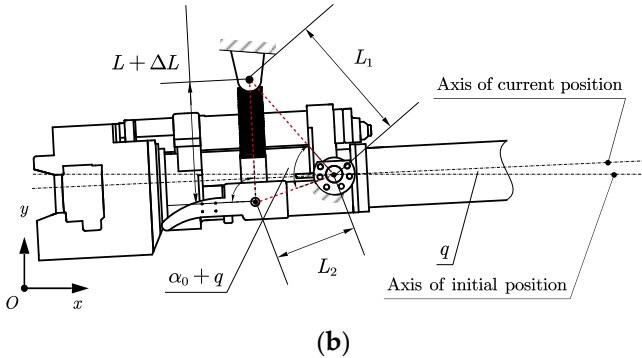

**(b)**

**Figure 1.** Geometric relationships: (**a**) Real-world tank; (**b**) Vehicle stabilization system.

Thus, the potential and kinetic energy of the tank vehicle stabilization system can be obtained as

$$
\begin{aligned}
E &= mD^2\dot{q}^2/6 \\
P &= mgD\sin q/2
\end{aligned}
\tag{1}
$$

where $m$ is the total mass, $D$ is the distance from the gravity center to the rotation center, and $g$ denotes the gravity acceleration of the tank vehicle stabilization system.

According to the kinetic function characterization of the Lagrange, there is

$$
K = E - P
\tag{2}
$$

Then according to Equation (2), the Euler-Lagrange equation can be obtained as

$$
\frac{\mathrm{d}}{\mathrm{d}t} \cdot \frac{\partial K}{\partial \dot{q}} - \frac{\partial K}{\partial q} = M
\tag{3}
$$

where $M$ denotes the driving torque that drives the tank vehicle stabilization system around the rotation center. Combined with Equation (1), one obtains

$$
M = \frac{1}{3}mD^2\ddot{q} + \frac{1}{2}mgD\cos q + d(t)
\tag{4}
$$

where $d(t)$ is the total uncertainty term, containing the uncertain nonlinearities, parametric nonlinearities, kinetic modeling errors, and unmodeled terms.

The whole vertical direction of the tank vehicle stabilization system is driven and controlled by a linear actuator. Specifically, the electric cylinder is used as the linear actuator in the vehicle stabilization system. The electric cylinder includes a permanent magnet synchronous motor (PMSM), reducer, and roller screw; its transmission principle can be expressed as Figure 2. Therefore, the dynamic balance equation inside the linear actuator is stated as

$$
\begin{cases}
J_a\ddot{\theta}_a + B_a\dot{\theta}_a + T_{av} = T_a \\
T_a = k_a u_a \\
J_v\ddot{\theta}_a + B_v\dot{\theta}_a + T_{bv} = T_{av} \\
T_{bv} = S_l(2\pi N)^{-1}F_e
\end{cases}
\tag{5}
$$

where $\theta_a$, $\dot{\theta}_a$ and $\ddot{\theta}_a$ denote the angular displacement, angular velocity, and angular acceleration of the linear actuator motor rotor. $J_a$ is the moment of inertia of the linear actuator motor, $B_a$ is the coefficient of friction of the linear actuator motor, $T_a$ is the electromagnetic torque of the linear actuator motor, $k_a$ is the torque coefficient of the linear actuator motor, $u_a$ is the control input of the linear actuator motor, $T_{av}$ is the output torque of the linear actuator motor, $T_{bv}$ is the output torque of the linear actuator through the reducer, $J_v$ is the moment of inertia of the actuator, $B_v$ is the friction coefficient of the actuator, $S_l$ is the lead of the screw, $F_e$ is the output force of the linear actuator, and $N$ is the ratio of the reducer.

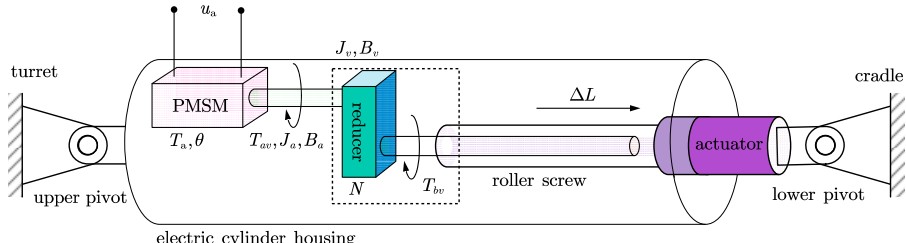

**Figure 2.** Internal structure of electric cylinder.

To simplify the arithmetic process, define the equivalent moment of inertia $J_{\text{equ}} \triangleq J_a + J_v$, the equivalent friction coefficient $B_{\text{equ}} \triangleq B_a + B_v$, and the transmission coefficient $A \triangleq S_l(2\pi N)^{-1}$. From this, Equation (5) can be rewritten as

$$J_{\text{equ}}\ddot{\theta}_a + B_{\text{equ}}\dot{\theta}_a + AF_e = k_a u_a \tag{6}$$

According to the cosine theorem, it is observed that in Figure 2, in the triangle formed by the upper and lower pivot points of the linear actuator and the rotation center of the tank vehicle stabilization system, the following exists:

$$\begin{aligned}
\Delta L &= \left(L_1^2 + L_2^2 - 2L_1L_2\cos\alpha\right)^{1/2} - L_0 \\
&= \left(L_1^2 + L_2^2 - 2L_1L_2\cos(q+\alpha_0)\right)^{1/2} - L_0
\end{aligned} \tag{7}$$

The electric cylinder rotor angular displacement $\theta_a$ and the extension length of the linear actuator $\Delta L$ are related as $\theta_a = A^{-1}\Delta L$. The ideal relationship between $\theta_a$ and $q$ can be obtained as

$$\theta_a = A^{-1}\left[\left(L_1^2 + L_2^2 - 2L_1L_2\cos(q+\alpha_0)\right)^{1/2} - L_0\right] \tag{8}$$

Taking the first- and second-order derivatives regarding time for $\theta_a$, the angular velocity $\dot{\theta}_a$ and angular acceleration $\ddot{\theta}_a$ of the rotor of the linear actuator can be obtained as

$$\begin{aligned}
\dot{\theta}_a &= \frac{L_aL_d\sin(q+\alpha_0)\dot{q}}{A\left(L_a^2+L_d^2-2L_aL_d\cos(q+\alpha_0)\right)^{1/2}} \\
\ddot{\theta}_a &= \frac{L_aL_d\cos(q+\alpha_0)\dot{q}^2+L_aL_d\sin(q+\alpha_0)\ddot{q}}{A\left(L_a^2+L_d^2-2L_aL_d\cos(q+\alpha_0)\right)^{1/2}} \\
&\quad - \frac{L_a^2L_d^2\sin^2(q+\alpha_0)^2\dot{q}}{A\left(L_a^2+L_d^2-2L_aL_d\cos(q+\alpha_0)\right)^{3/2}}
\end{aligned} \tag{9}$$

The dynamic balancing equation for the tank vehicle stabilization system is given as

$$M - F_eL_d\sin(q+\alpha_0) = 0 \tag{10}$$

Combining Equations (4), (6), and (10) yields

$$\begin{aligned}
u_a &= \frac{\left(L_1^2+L_2^2-2L_1L_2\cos(q+\alpha_0)\right)^{1/2}}{AL_1L_2\sin(q+\alpha_0)k_a}\left[\frac{1}{3}mD^2 + \frac{A^2L_1^2L_2^2J_{\text{equ}}\sin^2(q+\alpha_0)}{L_1^2+L_2^2-2L_1L_2\cos(q+\alpha_0)}\right]\ddot{q} \\
&\quad + \frac{AL_1L_2J_{\text{equ}}}{k_a\left(L_1^2+L_2^2-2L_1L_2\cos(q+\alpha_0)\right)^{1/2}}\left[\sin(q+\alpha_0) + \cos(q+\alpha_0)\dot{q}\right. \\
&\quad \left. - \frac{L_1L_2\sin^2(q+\alpha_0)}{L_1^2+L_2^2-2L_1L_2\cos(q+\alpha_0)}\dot{q}\right]\dot{q} \\
&\quad + \frac{\left(L_1^2+L_2^2-2L_1L_2\cos(q+\alpha_0)\right)^{1/2}mgD\cos q}{2k_aAL_1L_2\sin(q+\alpha_0)} \\
&\quad + \frac{\left(L_1^2+L_2^2-2L_1L_2\cos(q+\alpha_0)\right)^{1/2}}{AL_1L_2\sin(q+\alpha_0)k_a}d(t)
\end{aligned} \tag{11}$$

According to Equation (11), the final kinetic equation established in this part is

$$A(q)\ddot{q} + B(q,\dot{q})\dot{q} + C(q) + d'(t) = u_a \tag{12}$$

where $A(q)$, $B(q,\dot{q})$, $C(q)$, and $d'(t)$ are defined as

$$
\begin{aligned}
A(q) &= \frac{\left(L_1^2+L_2^2-2L_1L_2\cos(q+\alpha_0)\right)^{1/2}}{AL_1L_2\sin(q+\alpha_0)k_a}\left[\frac{1}{3}mD^2 + \frac{A^2L_1^2L_2^2J_{\mathrm{equ}}\sin^2(q+\alpha_0)}{L_1^2+L_2^2-2L_1L_2\cos(q+\alpha_0)}\right],\\
B(q,\dot{q}) &= \frac{AL_1L_2J_{\mathrm{equ}}}{k_a\left(L_1^2+L_2^2-2L_1L_2\cos(q+\alpha_0)\right)^{1/2}}\cdot\\
&\quad\left[\sin(q+\alpha_0) + \cos(q+\alpha_0)\dot{q} - \frac{L_1L_2\sin^2(q+\alpha_0)}{L_1^2+L_2^2-2L_1L_2\cos(q+\alpha_0)}\dot{q}\right],\\
C(q) &= \frac{\left(L_1^2+L_2^2-2L_1L_2\cos(q+\alpha_0)\right)^{1/2}mgD\cos q}{2k_aAL_1L_2\sin(q+\alpha_0)},\\
d'(t) &= \frac{\left(L_1^2+L_2^2-2L_1L_2cos(q+\alpha_0)\right)^{1/2}}{AL_1L_2\sin(q+\alpha_0)k_a}d(t).
\end{aligned}
\tag{13}
$$

## 3. Controller Design

The control objective of this paper is to design the control input $u_a$ such that the actual motion trajectory q of the vehicle stabilization system tracks the desired trajectory $q_{\mathrm{d}}$ and to ensure that all signals of the system are bounded. To facilitate the design of the controller, the following reasonable assumptions are given in this paper:

**Assumption 1.** *The reference instruction $q_{\mathrm{d}}$ of the tank vehicle stabilization system given in this paper is second-order continuously differentiable, and the first and second derivatives are bounded.*

**Assumption 2.** *The modeling errors and the unmodeled term of the tank vehicle stabilization system are continuous and bounded, and its first and second derivatives are also continuous and bounded.*

### 3.1. Sliding Mode Strategy

In this section, in order to ensure the effectiveness of the proposed nonlinear robust controller, a model-based sliding mode control strategy is designed for the vehicle stabilization system.

In the first step, the error variable *e* is defined as the deviation of the desired angular displacement $q_{\mathrm{d}}$ from the actual angular displacement *q*, that is,

$$e = q_{\mathrm{d}} - q \tag{14}$$

The sliding mode function is chosen as

$$s = \dot{e} + ce = 0 \tag{15}$$

where $c > 0$ is a design parameter. The sliding mode dynamic properties ensure the stability of the system and the convergence of the states to the equilibrium state $e = 0$. Combining Equation (14) and derivatizing for *s* yields

$$
\begin{aligned}
s &= \dot{q}_{\mathrm{d}} - \dot{q} + ce\\
\dot{s} &= \ddot{q}_{\mathrm{d}} - \ddot{q} + c\dot{e}
\end{aligned}
\tag{16}
$$

Combining Equations (12) and (16), one obtains

$$
\begin{aligned}
\dot{s} &= \ddot{q}_{\mathrm{d}} + c\dot{e} + A^{-1}\left(-u_a + B\dot{q} + C + d'\right)\\
&= \ddot{q}_{\mathrm{d}} + c\dot{e} + A^{-1}\left[-u_a + B\left(\dot{q}_{\mathrm{d}} + ce - s\right) + C + d'\right]\\
&= A^{-1}\left(-Bs - u + d' + f\right)
\end{aligned}
\tag{17}
$$

where $f = A\left(\ddot{q}_{\mathrm{d}} + c\dot{e}\right) + B\left(\dot{q} + ce\right) + C$.

The sliding mode control has the advantages of fast response and insensitivity to disturbances. However, due to its own characteristics, sliding mode control has the phenomenon of jitter, and this phenomenon cannot be eliminated by the control system. By choosing a suitable convergence law, it is possible to achieve a faster convergence to zero with the same control jitter or to reduce the jitter with the same convergence speed.

In order to cut down the jitter, this paper adopts the sliding mode function based on the exponential convergence law for

$$\dot{s} = -\varepsilon \mathrm{sgn}(s) - \gamma s \tag{18}$$

where $\varepsilon > 0$, $k > 0$, $\mathrm{sgn}(\cdot)$ is the sign function, $-ks$ is the exponential convergence term, and $-\varepsilon\mathrm{sgn}(s)$ is an isochronous convergence term that can be used to reduce jitter. Here, the larger k is, the faster $-\varepsilon\mathrm{sgn}(s)$ converges to the sliding mode surface. Therefore, increasing $k$ and decreasing $\varepsilon$ ensures fast convergence while reducing jitter.

### 3.2. Neural Network Estimation

The neural networks play a crucial role in control systems due to their powerful function approximation capabilities. Considering their ability to approximate general functions effectively, neural networks are well-suited for modeling and controlling complex dynamic systems with smooth nonlinear characteristics. By appropriately adjusting the weight parameters within three or more layers of neural networks, almost any smooth nonlinear function can be accurately approximated. This flexibility and adaptability make neural networks a valuable tool in the design and implementation of advanced control strategies, enabling the control system to effectively handle a wide range of nonlinearities and uncertainties in real-world applications. The three-layer neural network can be used to approximate the nonlinear functions $f$ and obtain

$$f(\boldsymbol{x}_u) = \boldsymbol{W}^{\mathrm{T}}\sigma(\boldsymbol{x}_u) + b_{\mathrm{approx}}(\boldsymbol{x}_u) \tag{19}$$

where $\boldsymbol{x}_u = [1, x_1, x_2, u_a]^{\mathrm{T}}$ is the neural network input, $W \in R^{H+1}$ is the ideal weight corresponding to the hidden layer to the output layer, $\sigma(\cdot)$ is the activation functions of the neural network, and $b_{\mathrm{approx}}(\cdot)$ is the approximation error of the neural network. It is worth noting that the approximation error $b_{\mathrm{approx}}(\boldsymbol{x}_u)$ of the neural network and its derivatives are bounded, i.e., they are

$$\left|b_{\mathrm{approx}}(\boldsymbol{x}_u)\right| \le \vartheta_1, \left|\dot{b}_{\mathrm{approx}}(\boldsymbol{x}_u)\right| \le \vartheta_2 \tag{20}$$

where $\vartheta_1$ and $\vartheta_2$ are known positive constants.

The biased neural network built above can be represented as shown in Figure 3.

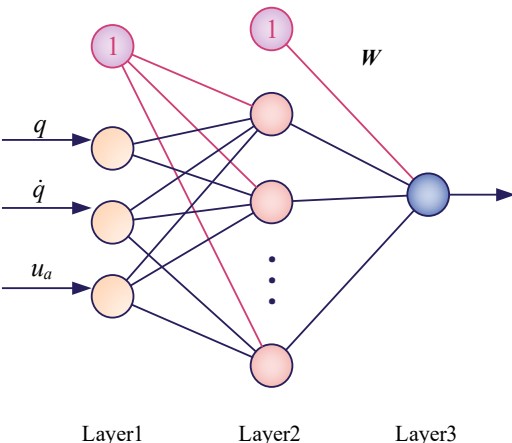

**Figure 3.** Three-layer feedforward neural network structure.

$\hat{f}$ is defined as the estimate of $f$ with

$$\hat{f}(\boldsymbol{x}_u) = \overset{\wedge}{\boldsymbol{W}}^{\mathrm{T}} \sigma(\boldsymbol{x}_u) \tag{21}$$

To ensure the stability of the adaptive law of neural network weights, the following discontinuous mapping is defined based on a bounded range of ideal weights:

$$proj_{\hat{W}_i}(\cdot_i) = \begin{cases} 0 & \hat{W}_i = \hat{W}_{i\max} \text{ and } \cdot_i > 0 \\ \cdot_i & \text{other} \\ 0 & \hat{W}_i = \hat{W}_{i\min} \text{ and } \cdot_i < 0 \end{cases} \tag{22}$$

where $\overset{\wedge}{\boldsymbol{W}}$ is the estimate of the ideal weight $\boldsymbol{W}$ of the neural network. The weight estimation error $\tilde{\boldsymbol{W}}$ is defined as

$$\tilde{\boldsymbol{W}} = \boldsymbol{W} - \overset{\wedge}{\boldsymbol{W}}. \tag{23}$$

To realize the adaptive approximation, the following neural network adaptive law is defined:

$$\dot{\overset{\wedge}{\boldsymbol{W}}} = proj_{\hat{W}_i}(\boldsymbol{\Gamma}\boldsymbol{\tau}), \ \boldsymbol{W}_{\min} \leq \overset{\wedge}{\boldsymbol{W}}(0) \leq \boldsymbol{W}_{\max} \tag{24}$$

where $\boldsymbol{\Gamma}$ is a positive definite diagonal matrix. For any adaptive rate $\boldsymbol{\tau}$, the discontinuous mapping makes the following equation hold:

$$\begin{cases} \overset{\wedge}{\boldsymbol{W}} \in \Omega_{\overset{\wedge}{\boldsymbol{W}}} \quad \overset{\wedge}{\boldsymbol{W}} : \boldsymbol{W}_{\min} \leq \overset{\wedge}{\boldsymbol{W}} \leq \boldsymbol{W}_{\max} \\ \tilde{\boldsymbol{W}}^{\mathrm{T}} \left[ \boldsymbol{\Gamma}^{-1} proj_{\hat{W}_i}(\boldsymbol{\Gamma}\boldsymbol{\tau}) - \boldsymbol{\tau} \right] \leq 0 \quad \forall \boldsymbol{\tau} \end{cases} \tag{25}$$

In this paper, $\boldsymbol{\tau} = \sigma(\boldsymbol{x}_u)s^{\mathrm{T}}$ is taken to guarantee the bounded stability of the final controller. Bringing Equation (24) into Equation (17) yields

$$\dot{s} = A^{-1}\left(-u + \hat{f} - Bs + d'\right) \tag{26}$$

Combining Equation (26) and Equation (18), the control input $u_a$ of the tank vehicle stabilization is given as

$$u_a = A(\varepsilon \mathrm{sgn}(s) + \gamma s) - Bs + \hat{f} - v \tag{27}$$

Here $v$ is the sliding mode robust term that forces the system to fluctuate near the sliding mode surface after first reaching the equilibrium point at $s = 0$, avoiding leaving the sliding mode surface due to the presence of the disturbance term $d'$. Its preliminary design is as

$$v = -\left(\varepsilon_N + d'_N\right)\mathrm{sgn}(s) \tag{28}$$

where $\varepsilon_N$ and $d'_N$ are design parameters which are well-designed to guarantee the asymptotic capability of the proposed controller.

Due to the discontinuity of the sign function $y = \mathrm{sgn}(x)$ at the origin, there is a phenomenon of oscillation when the system state reaches the equilibrium point. The oscillation phenomenon can cause a lot of harmful effects in applications, such as extra energy consumption and heat generation. Therefore, the robust term must be refined, and the key to refinement lies in changing the continuity of the robust term at the origin. Hence, the function $y = \tanh(x/\xi)$ is introduced as

$$\tanh(x/\xi) = \frac{e^{x/\xi} - e^{-x/\xi}}{e^{x/\xi} + e^{-x/\xi}} \tag{29}$$

where $0 < \xi \leq 1$ is a design parameter. The correspondence of this function with $y = \text{sgn}(x)$ at the origin is shown in Figure 4. It can be seen that the function is significantly improved in terms of continuity and smoothness at the origin compared to $y = \text{sgn}(x)$.

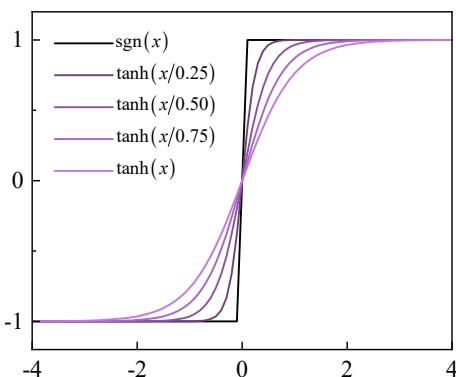

**Figure 4.** Comparison of function $y = \text{sgn}(x)$ and function $y = \tanh(x/\xi)$.

Therefore, the sliding mode robust term $v$ is redesigned as

$$v = -\left(\varepsilon_N + d'_N\right)\tanh(x/\xi) \tag{30}$$

*3.3. Stability Analysis*

The Lyapunov function can be selected as

$$L = \frac{1}{2}s^{\mathrm{T}}As + \frac{1}{2}\text{tr}\left(\tilde{\boldsymbol{W}}^{\mathrm{T}}\boldsymbol{\Gamma}^{-1}\tilde{\boldsymbol{W}}\right) \tag{31}$$

Taking the derivative for $V_L$ and combining Equation (26), we obtain

$$
\begin{aligned}
\dot{L} &= s^{\mathrm{T}}A\dot{s} + \tfrac{1}{2}s^{\mathrm{T}}\dot{A}s + \text{tr}\left(\tilde{\boldsymbol{W}}^{\mathrm{T}}\boldsymbol{\Gamma}^{-1}\dot{\tilde{\boldsymbol{W}}}\right) \\
&\leq s^{\mathrm{T}}A(-\varepsilon\,\text{sgn}(s) - \gamma s) + s^{\mathrm{T}}(\varepsilon + v + d) \\
&\quad + \tilde{\boldsymbol{W}}^{\mathrm{T}}\sigma s^{\mathrm{T}} + \text{tr}\left(\tilde{\boldsymbol{W}}^{\mathrm{T}}\boldsymbol{\Gamma}^{-1}\dot{\tilde{\boldsymbol{W}}}\right)
\end{aligned}
\tag{32}
$$

**Lemma 1.** *From related knowledge of linear algebra, when both* $\boldsymbol{a}$ *and* $\boldsymbol{b}$ *are n-dimensional column vectors, their trace operations have the following properties:*

$$\text{tr}\left(\boldsymbol{a}^{\mathrm{T}}\boldsymbol{b}\right) = \boldsymbol{b}\boldsymbol{a}^{\mathrm{T}} \tag{33}$$

According to Lemma 1, one obtains

$$
\begin{aligned}
\dot{L} &= s^{\mathrm{T}}A(-\varepsilon\,\text{sgn}(s) - \gamma s) + s^{\mathrm{T}}(\varepsilon + v + d') \\
&\quad + \tilde{\boldsymbol{W}}^{\mathrm{T}}\sigma s^{\mathrm{T}} + \boldsymbol{\Gamma}^{-1}\dot{\tilde{\boldsymbol{W}}}\tilde{\boldsymbol{W}}^{\mathrm{T}} \\
&= s^{\mathrm{T}}A(-\varepsilon\,\text{sgn}(s) - \gamma s) + s^{\mathrm{T}}(\varepsilon + v + d') \\
&\quad + \text{tr}\left[\tilde{\boldsymbol{W}}^{\mathrm{T}}\left(\sigma s^{\mathrm{T}} + \boldsymbol{\Gamma}^{-1}\dot{\tilde{\boldsymbol{W}}}\right)\right] \\
&\leq s^{\mathrm{T}}(\varepsilon + v + d')
\end{aligned}
\tag{34}
$$

Considering the definition of the robust term $v$ in Equation (28), it is only necessary that the design parameters satisfy that, at any moment, $\varepsilon_N + d'_N \geq \varepsilon + d'$, then $\dot{L} < 0$. Since

$L$ is positive definite, the tank vehicle stabilization system can achieve asymptotic stability performance. At this point, the proof of stability for the presented controller is completed.

## 4. Co-Simulation Verification

### 4.1. Co-Imulation Principle

This part validates the proposed control algorithm utilizing co-simulation. The roadway is modeled with different grades of roughness according to ISO standards [31]. The main structure of the tank is analyzed and the multi-body dynamics of the tank is modeled in RecurDyn software (version V9R2). Meanwhile, the comparison controllers are built in MATLAB/Simulink (version R2023b). The Sensor function in RecurDyn can be utilized to read the motion data of the gun muzzle in real time and send it back to the controller in MATLAB/Simulink. The control system in MATLAB/Simulink outputs the PMSM control voltage of the vehicle stabilization system in real time, thus realizing the co-simulation of the closed-loop process. The principle of the co-simulation is illustrated in Figure 5.

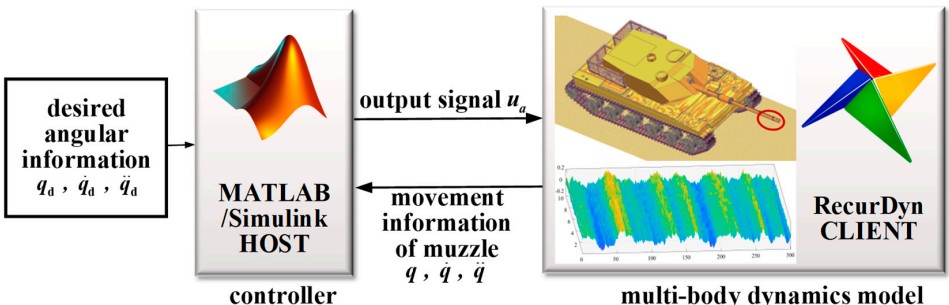

**Figure 5.** Principle of co-simulation.

### 4.2. Model Development

4.2.1. The Establishment of Pavement Spectrum

The primary factor causing the tank vehicle stabilization system to fluctuate in the vertical direction is pavement excitation. As the tank moves, pavement excitation will be transmitted to the hull, turret, and barrel via the tracks and suspension system. This will lead to forcing vibration at the barrel, which affects the projectile attitude as it leaves the muzzle, thus reducing strike accuracy.

The signal of pavement roughness is a stationary Gaussian process. The power spectral density typically describes its statistical characteristics. According to pavement surface profile classification, the power spectral density of pavement roughness [32] can be calculated as

$$G_q(n) = G_q(n_0)\left(\frac{n}{n_0}\right)^{-w} \tag{35}$$

where $n = 1/\lambda$ is the spatial frequency and $\lambda$ is the wavelength; $n_0$ is the reference spatial frequency; $G_q(n_0)$ is the pavement unevenness coefficient, which characterizes the pavement power spectral density value at the reference spatial frequency $n_0$; and $w$ is the frequency index, which is the slope of the slope line on the double logarithmic coordinate. The frequency composition of the pavement power spectral density defines the classification of pavement roughness into eight grades, denoted as A through H, based on the characteristics observed in the pavement power density spectrum.

Dividing the pavement spatial frequency $f(f_1 < f < f_2)$ into $N$ intervals and taking the power spectral density value $G_q(f_i)$ corresponding to the central frequency $f_i(i = 1, 2, \ldots, N)$ of each interval as an approximate substitute for the value of the entire interval range $\Delta n$, the pavement roughness [7,8] can be written as

$$q(x) = \sum_{i=1}^{N}\left\{\sqrt{2}A_i \cdot \sin[2\pi(n_i x + \alpha_i)]\right\} \tag{36}$$

where the $x$ direction is the direction along the pavement surface, $\alpha_i$ is the random number uniformly distributed in $[0,1]$, and $A_i$ is the vibration amplitude of the harmonic corresponding to the central frequency $f_i$, denoted as

$$A_i = \sqrt{G_q(f_i) \cdot \Delta n} \tag{37}$$

In addition, in the actual moving process, the excitation from the pavement is not the same on both sides of the tank tracks, and the coherence is be expressed as

$$\gamma(n) = \begin{cases} e^{-\rho n d_v} & n \in (n_1, n_2) \\ 0 & n \notin (n_1, n_2) \end{cases} \tag{38}$$

where $d_v$ is the wheelbase, $\rho$ is the empirical value, and $n_1$ and $n_2$ are the upper and lower limits of the spatial frequency of the roadway, respectively. Since the stochastic phase angle $\alpha_{Ri}$ in the pavement unevenness function is the main cause of the difference in track excitation between the two sides, depending on Equation (7), the phase angle coherence of the tracks on both sides under pavement excitation can be written as

$$\alpha_{Ri} = \frac{e^{-2\pi d_v n^{1.5}} \alpha_i + \sqrt{1 - e^{-2\pi x n^{1.5}}} \alpha_n}{\sqrt{1 - e^{-2\pi d_v n^{1.5}}} + e^{-2\pi d_v n^{1.5}}} \tag{39}$$

where $\alpha_n$ is the freshly generated random number within $[0, 1]$.

Combining Equations (36) and (39), the three-dimensional pavement unevenness stochastic process is written as

$$q(x, y) = \sum_{i=1}^{N} \left\{ \sqrt{2} A_i \cdot \sin\left[2\pi\left(n_i x + \alpha_y\right)\right] \right\} \tag{40}$$

and

$$\alpha_y = \frac{e^{-2\pi y n^{1.5}} \alpha_i + \sqrt{1 - e^{-2\pi y n^{1.5}}} \alpha_n}{\sqrt{1 - e^{-2\pi y n^{1.5}}} + e^{-2\pi y n^{1.5}}} \tag{41}$$

where $\alpha_y$ is the random phase for the $y$ direction and the $y$ direction is the direction perpendicular to the pavement.

Using the harmonic superposition method to reconstruct the pavement, a random generation program of pavement roughness is written based on mathematical tool. E-grade pavement with a length of 300 m and a width of 10 m are established, and a software-readable pavement file is created as shown in Figure 6.

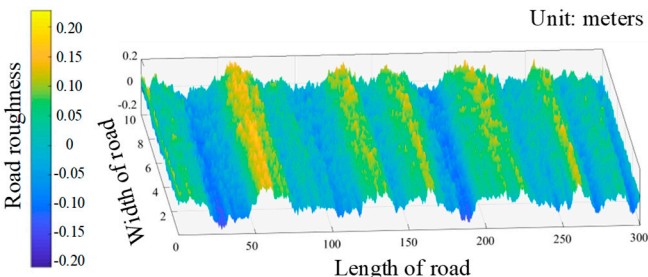

**Figure 6.** E-grade pavement profile in a three-dimensional plot.

4.2.2. The Multi-Body Dynamics Model of Moving Tanks

The multi-body dynamics of the tank vehicle stabilization system mainly includes five units: the track unit, hull unit, turret unit, elevating unit, and recoil unit. It also involves the constraint connection between assemblies, the establishment of assembly positioning and pavement contact pairs, and the solution of large-scale complex structure contact problems.

The assembly and connection relationship mainly considered in this paper are as Figure 7.

**Figure 7.** Topological relationship of tanks.

For fully considering contact collision and other uncertain disturbances in the actual moving tank system, a multi-body dynamics model of the tank system is created in Recur-Dyn. The primary components considered in tank modeling are those listed in the previous section, as shown in Figure 8.

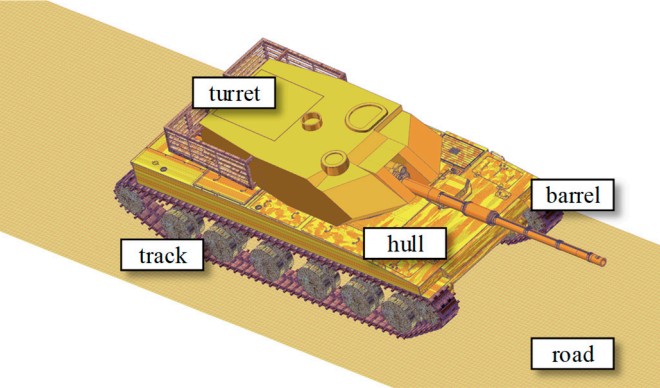

**Figure 8.** Multi-body dynamics model of moving tank.

In the software, it is possible to set different initial tank speeds. It is also possible to make the unevenness of the pavement surface change by changing $G_q(n_0)$. This model is used for simulation, which can initially verify the feasibility and superiority of the proposed controller.

### 4.3. Co-Simulation Results

In this paper, two cases of simulation experiments are implemented. Case 1 is a fixed position tracking simulation, and Case 2 is a dynamic position tracking simulation.

The simulation process for Case 1 is as follows. The total simulation duration is set to 10 s. According to the physical factors of the actual tank gun angle-pointing system, the maximum angular velocity is no more than 0.436 rad $\cdot$ s$^{-1}$ and the maximum angular

acceleration is no more than $0.698 \text{ rad} \cdot \text{s}^{-2}$. The desired position command for the angular displacement of the muzzle is given as $x_{1d} = 0.1\left(1 - e^{-0.5t^3}\right)$, as shown in Figure 9. The fixed angular displacement is 0.1 rad.

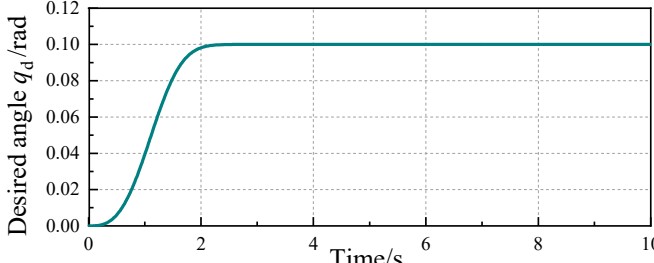

**Figure 9.** Case 1: desired fixed-position angle command.

The simulation process for Case 2 is as follows. To verify the dynamic tracking performance of the proposed controller, given the desired control sinusoidal position command $x_{1d} = 0.05\left(1 - e^{-0.001t^3}\right)\sin 0.25\pi t$. The limits of movement are the same as in Case 1. The total simulation duration is set to 40 s, which is shown in Figure 10.

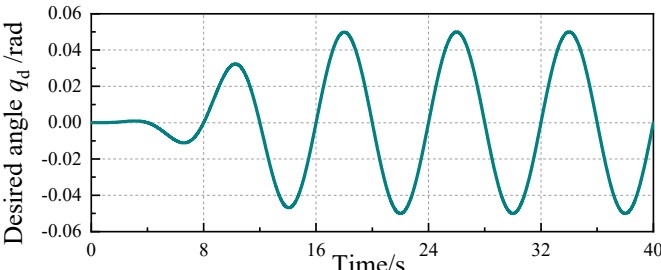

**Figure 10.** Case 2: desired dynamic-position angle command.

The dynamics model of the moving tank established above is used for simulation jointly with different controllers. The driving situation is a tank moving with a speed of 30 km/h on the E-grade pavement. The pavement parameters, dynamics parameters, and motor parameters involved are presented in Table 1.

**Table 1.** Major physical parameters within co-simulation.

| Parameters | Values | Units |
|:---:|:---:|:---:|
| $L_1$ | 0.440 | m |
| $L_2$ | 0.320 | m |
| $L_0$ | 0.400 | m |
| $\alpha_0$ | 1.07 | rad |
| $m$ | 2048.0 | kg |
| $S_l$ | 20 | mm |
| $N$ | 5 | - |
| $B_{equ}$ | $1.00 \times 10^{-3}$ | $N \cdot m \cdot s/rad$ |
| $J_{equ}$ | 0.100 | $kg \cdot m^2$ |
| $k_a$ | $1.95 \times 10^{-1}$ | $N \cdot m/A$ |
| $n_0$ | 0.1 | $m^{-1}$ |
| $w$ | 2 | - |
| $G_q(n_0)$ | $2.048 \times 10^{-3}$ | E-grade |

To clearly demonstrate the effectiveness and superiority of the proposed controller, two other existing controller strategies are selected for comparison. The three controllers for the comparison are the following:

SMCNN: This is the neural network-based sliding mode controller proposed in this paper. In robust terms, $\varepsilon_N = 0.55$ and $d'_N = 1.00$. Meanwhile, let $c = 0.3$. The neural network weight gain is taken as $\Gamma = 80$. The number of hidden layers in both neural networks is set to 10, while the activation function is taken as the hyperbolic tangent function, given as

$$\sigma(x) = -1 + \frac{2}{1 + e^{-2x}} \tag{42}$$

SMC: This is a conventional sliding mode controller that does not use neural networks for uncertainty compensation. By performing comparisons with SMC, we can verify the validity of the neural network to estimate the uncertainty of the vehicle stabilization system and achieve the feedforward compensation. The involved parameters are the same as the SMCNN described above.

PID: It is a proportional–integral–derivative controller, which is set by three parameters, $K_p$, $K_i$, and $K_d$. The control parameters are chosen as $K_p = 1300$, $K_i = 600$, and $K_d = 12$. The angle tracking results of the above two cases are shown in Figures 11 and 12.

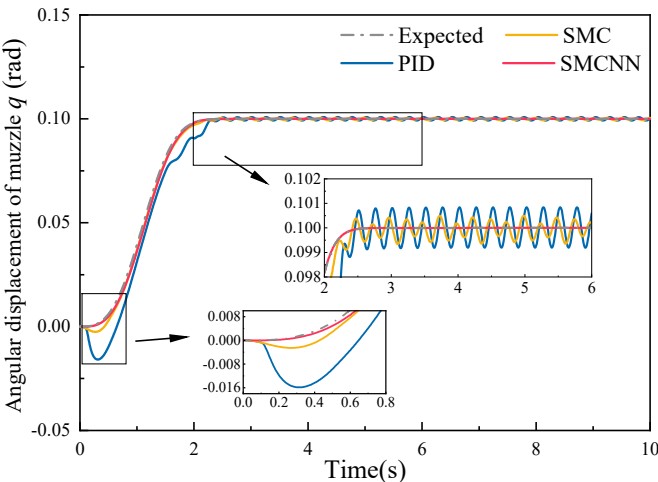

**Figure 11.** Comparison of angular positions for Case 1.

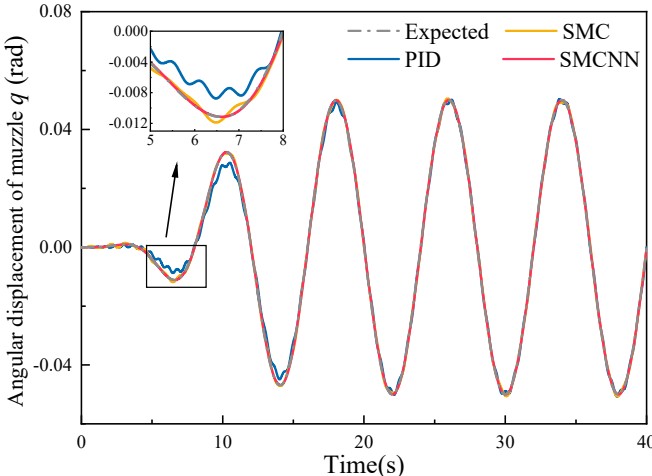

**Figure 12.** Comparison of angular positions for Case 2.

By observing the comparison of the control effects in case 1, it can be found that in the initial motion phase of the vehicle stabilization system, the SMCNN controller proposed in this paper can obtain a faster convergence speed and better tracking performance compared with the other two controllers. When the time surpasses 0.9 s, the tracking error of the SMCNN controller is less than 1% and the vehicle stabilization system reaches a relatively

stable stage. The time for the PID controller and the traditional SMC controller to reach a relatively stable stage is 2.6 s and 1.3 s, respectively, which illustrates the effectiveness of the combination of the sliding mode controller and the neural network controller, as was the case in this paper, when applied to tank vehicle stabilization systems in high-mobility environments. The adverse effects of unmodeled perturbations are reduced by the effective estimation of system uncertainty by neural network, and the design of sliding mode robust feedback control law improves the robustness of the vehicle stabilization system in complex environments, which ensures an excellent control effect in the SMCNN control method. After the control command reaches 0.1 rad, both SMC and PID have different degrees of deviation, while SMCNN still performs stably.

Through the analysis of case 2, it can be clearly found that the SMCNN controller proposed in this paper has better dynamic tracking effect than the traditional sliding mode SMC controller and PID controller. It lays a foundation for the vehicle stabilization system to achieve movement-by-movement attack by tank guns during their operation. As can be seen from Figure 12, the SMCNN controller proposed in this paper can maintain an excellent tracking effect within 0–7 s of the first command peak, while the convergence speed of the SMC controller is significantly lower than that of the SMCNN controller. At the same time, the tracking error of the SMC controller is clearly greater than that of SMCNN controller within 0–7 s. This further shows that the neural network controller proposed in this paper has excellent learning performance and accurate compensation for system uncertainty. When the time surpasses 20 s, the three contrast controllers can track the given expected motion instruction, but the SMCNN controller still ensures that the system obtains the minimum tracking error, especially at the peak of each sine instruction; the superiority of SMCNN is more clear. This again fully demonstrates the effectiveness of the nonlinear robust controller proposed in this paper by combining sliding mode control with a neural network. The traditional PID controller still has the worst tracking performance among the three comparison controllers.

The control inputs for the two cases are given in Figures 13 and 14. The control input indirectly indicates the control spend of the controller. Lower crests and smaller frequencies result in smaller transient torques of the linear actuator. It can be seen that the proposed SMCNN has the smallest control cost, which has great significance for engineering applications.

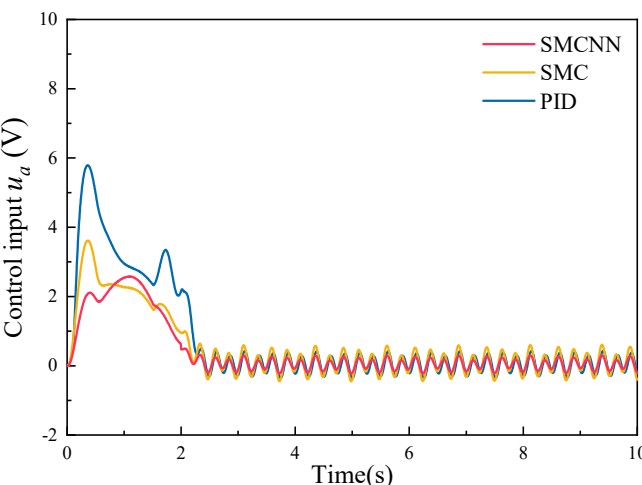

**Figure 13.** Comparison of control input for Case 1 in co-simulation verification.

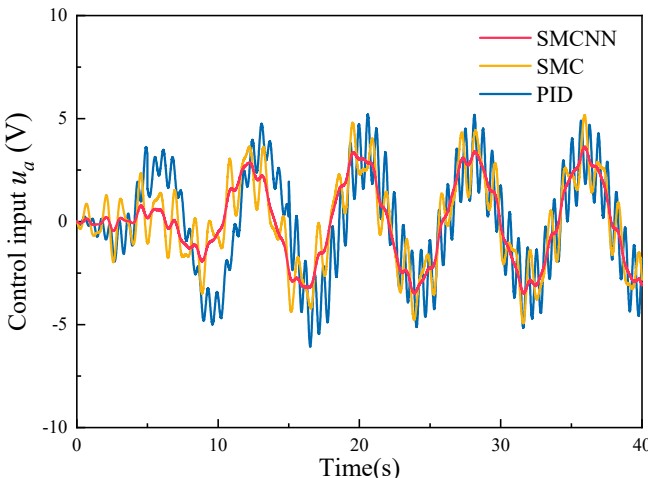

**Figure 14.** Comparison of control input for Case 2 in co-simulation verification.

Furthermore, in order to demonstrate the robustness of the closed-loop system to parameter variations, co-simulations with different parameter variations are performed. Since the total mass parameter is the most likely to vary in real-world conditions, it is thus selected for study. Specifically, in addition to the standard experiment with a nominal mass of 2048 kg, comparisons with 1998 kg and 2098 kg are carried out. The Figure 15 shows the co-simulation results. It can be noticed that the controller still performs well even when the total mass is changed. The control effect of the controller after a change in total mass is very close to the control effect at a nominal mass of 2048 kg. This is beneficial in the complex and changing battlefield environment. In addition, it can be found that the control effect is best when the mass is 2048 kg, which further illustrates the effectiveness of the model-based control strategy design. If the modeling is more accurate, the control performance of the model is also better.

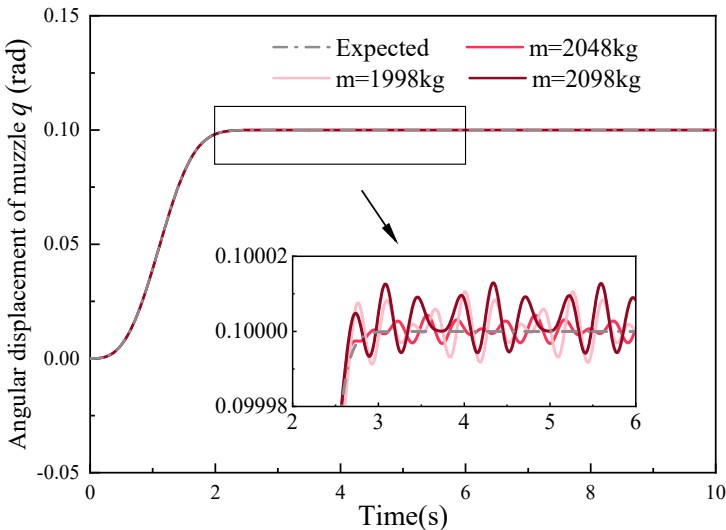

**Figure 15.** Comparison of different total masses.

To demonstrate the dynamic control performance of the proposed controller at different frequencies, co-simulations at different dynamic-position angle commands are required. Consequently, the co-simulations under control commands $x_{1d} = 0.05\left(1 - e^{-0.001t^3}\right)\sin 0.20\pi t$ and $x_{1d} = 0.05\left(1 - e^{-0.001t^3}\right)\sin 0.30\pi t$ are carried out, which are used to compare the control results with Figure 12. The angular displacements of the muzzle under these two control commands are demonstrated in Figures 16 and 17. Comparing these with Figure 12, it can

be found that the proposed controller exhibits excellent tracking performance at different dynamic frequencies under different dynamic control instructions. Moreover, as the frequency increases, the performance is even better compared to the other two controllers. This provides theoretical validation of the dynamic performance of the proposed algorithm in a real-world environment.

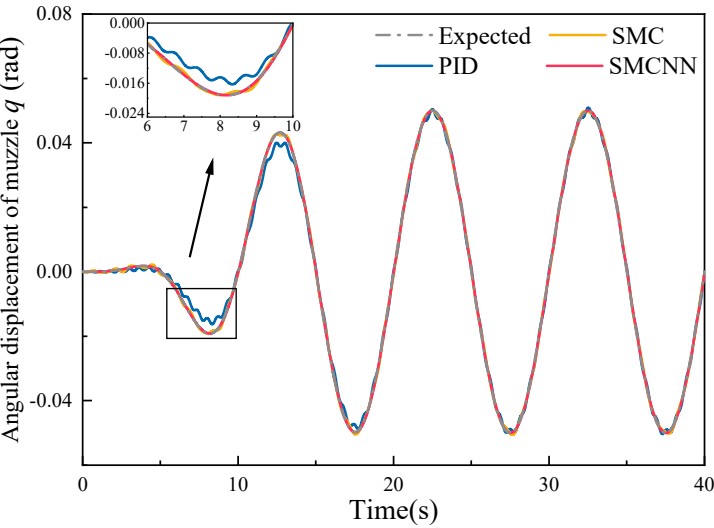

**Figure 16.** Comparison of angular positions for $x_{1d} = 0.05\left(1 - e^{-0.001t^3}\right)\sin 0.20\pi t$.

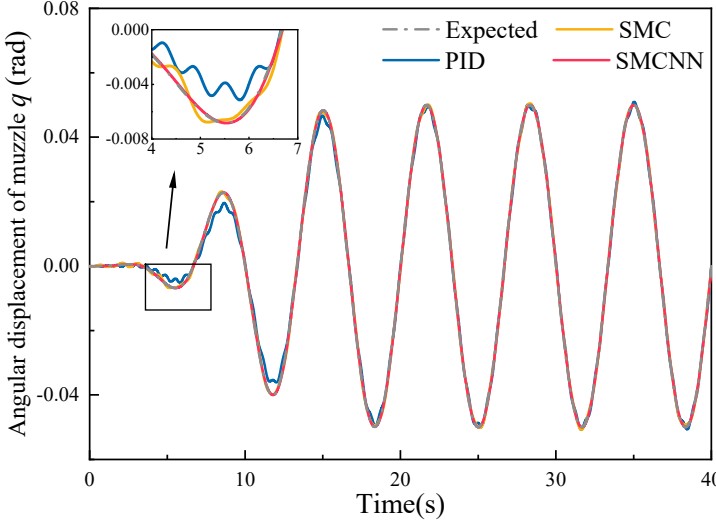

**Figure 17.** Comparison of angular positions for $x_{1d} = 0.05\left(1 - e^{-0.001t^3}\right)\sin 0.30\pi t$.

## 5. Experimental Verification

### 5.1. Composition Introduction

In this paper, an experimental platform for the tank vehicle stabilization system is constructed based on a specific type of fully electric tank gun with scaling theory. It is mainly composed of a mechanical part, electrical part, and testing part. Its overall appearance is shown in Figure 18.

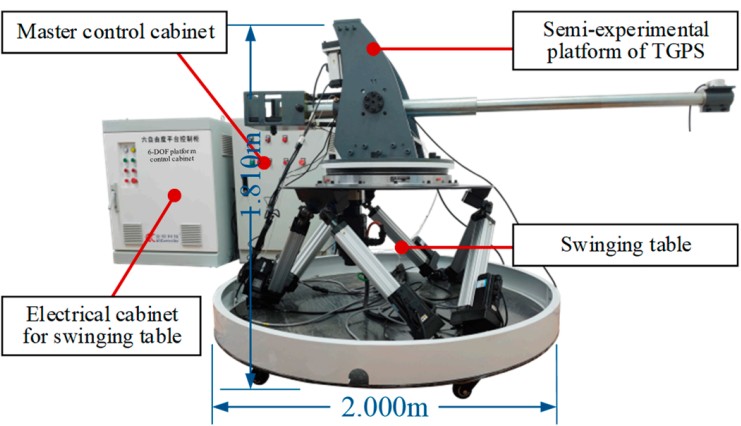

**Figure 18.** Overall appearance of the experimental platform.

The electrical part of the experimental platform is electrically controlled by means of servo motors driving linear actuators to achieve the pitching motion of the stabilized tank gun barrel. Electronic control equipment mainly controls the pitching motion of the test stand and its parameter setting. The system processor used is a TMS320F28335 DSP with a main frequency of 150 MHz. After the parameters such as the working mode, desired angle, and desired angular speed are set through the HMI control interface, the electrical equipment in the test stand starts to work according to the parameter settings, completes the pitch motion function, and the control system moves according to the given parameters.

The shaking table is utilized to simulate the excitation caused by pavement roughness. The gun breech has an adjustable weight design. Each external interface adopts a common interface, while relevant sensors are installed in the muzzle, cradle, and other locations. The console also integrates a certain number of network interfaces, asynchronous serial interfaces, and image interfaces, which are used to perform the data exchange and information transmission in the test stand. Figure 19 shows the sensors and main components of the part of the tank vehicle stabilization system. Figure 20 shows the principle of the experiment.

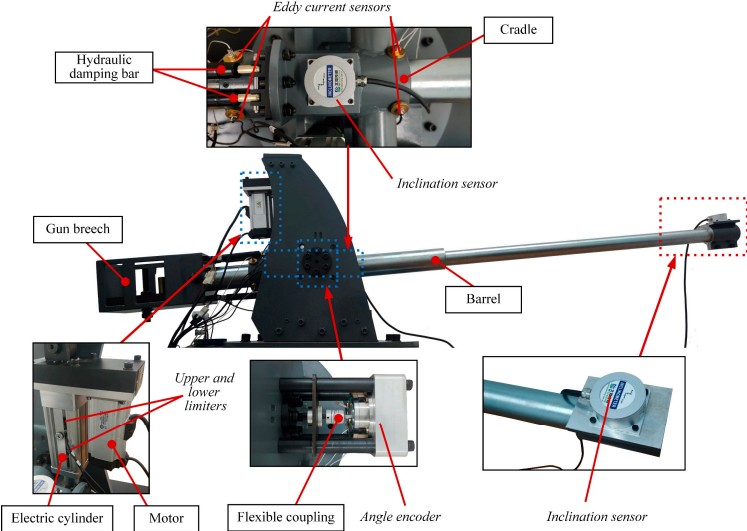

**Figure 19.** Parts of the tank vehicle stabilization system.

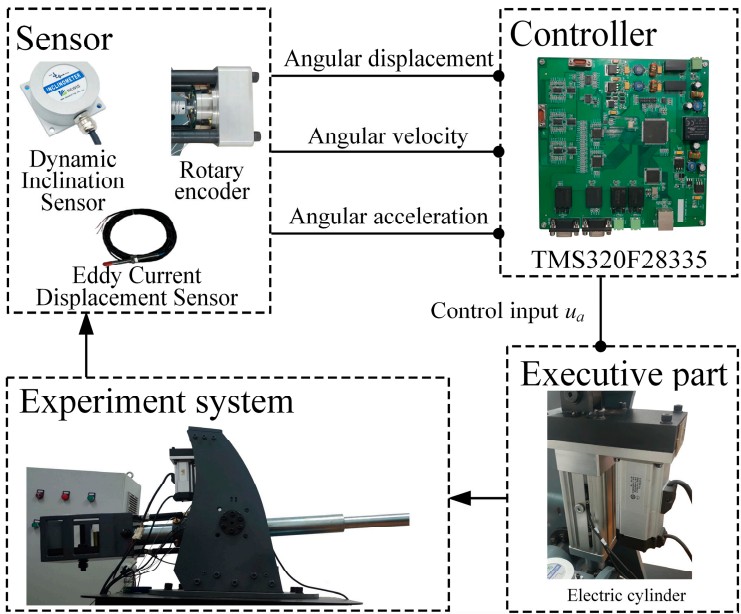

**Figure 20.** Principle of the experiment.

The dynamic parameters of the experimental platform involved, as well as the parameters of the linear actuator involved, are presented in Table 2.

**Table 2.** Major physical parameters within experimental platform.

| Parameters | Values | Units |
|:---:|:---:|:---:|
| $L_1$ | 0.194 | m |
| $L_2$ | 0.170 | m |
| $L_0$ | 0.094 | m |
| $\alpha_0$ | 0.50 | rad |
| $m$ | 71.01 | kg |
| $S_l$ | 6 | mm |
| $N$ | 10 | - |
| $B_{\text{equ}}$ | $1.00 \times 10^{-3}$ | $\text{N} \cdot \text{m} \cdot \text{s/rad}$ |
| $J_{\text{equ}}$ | 0.010 | $\text{kg} \cdot \text{m}^2$ |
| $k_a$ | $1.95 \times 10^{-1}$ | $\text{N} \cdot \text{m/A}$ |

*5.2. Experimental Results*

In the experiment, the driving speed of the tank is set to 30 km/h, and the roughness of the pavement is E-grade. Fixed-position tracking experiments are conducted. The angular position tracking responses of the three controllers are shown in Figure 21 when the control commands for Case 1 are input to the three controllers. Meanwhile, the angular tracking errors of the three controllers within 10 s can be obtained, as shown in the Figure 22. It can be observed that the proposed controller enters the stabilized control phase first. Moreover, after reaching the stabilized control phase, there are smaller control fluctuations compared to the other two controllers. This is thanks to the ability of the proposed controller to adjust the control parameters in real time to adapt to system changes and effectively resist external interference, as well as to have a faster response time and higher control accuracy. Intelligent features also bring more possibilities for system control. The estimate of the neural network $f$ for the SMCNN controller is shown in Figure 23. The control input is shown in Figure 24; it is continuous and bounded.

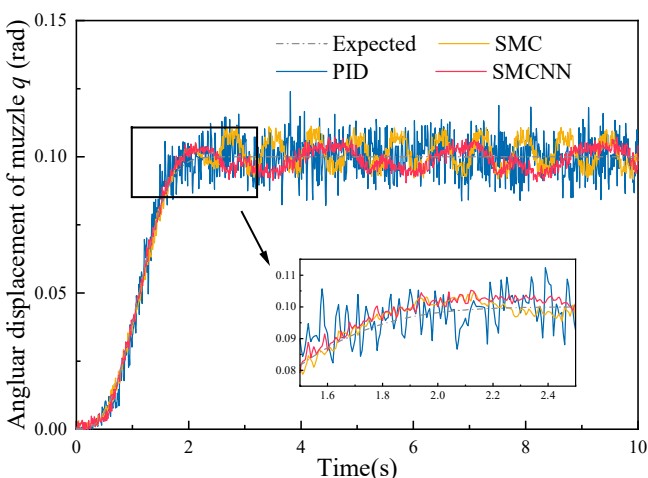

**Figure 21.** Fixed-position angle tracking results.

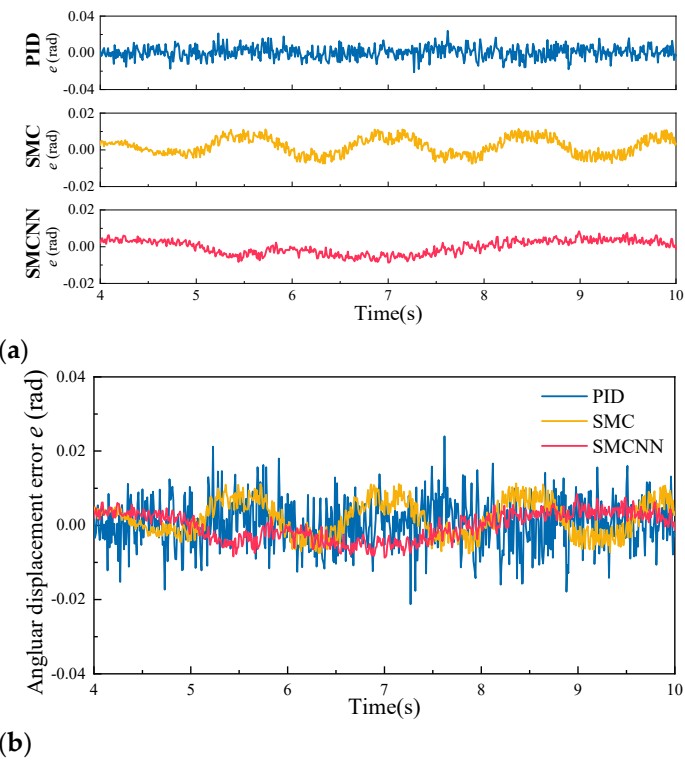

**Figure 22.** Fixed-position angle tracking error: (**a**) errors of the three controllers; (**b**) comparison.

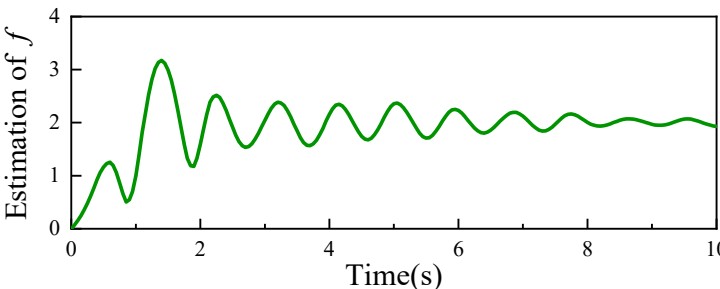

**Figure 23.** Neural network estimate $f$.

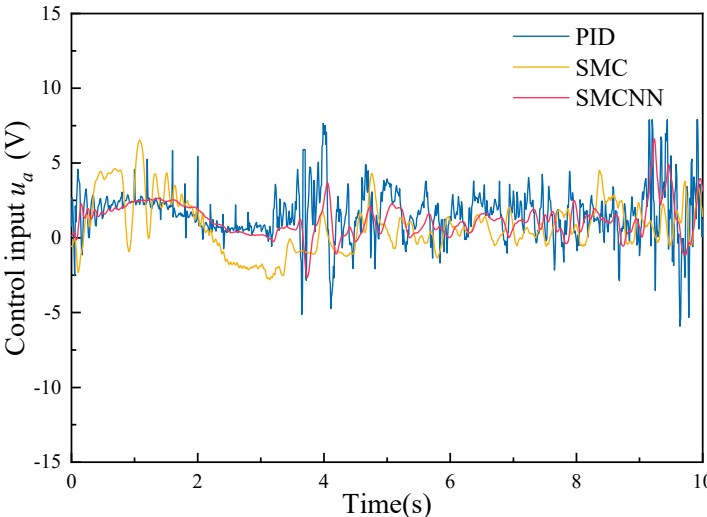

**Figure 24.** Comparison of control input for Case 1 in experimental verification.

Afterwards, dynamic-position tracking experiments are conducted. The dynamic position response of the controller designed in this paper can be obtained as seen in Figure 25. Dynamic errors of the controllers over the 40 s test time are obtained, as illustrated in Figure 26. The control input is shown in Figure 27, which is also continuous and bounded. The proposed controller shows more advantages in dynamic-position tracking experiments. It always tracks faster than the other two controllers. This is thanks to the neural network estimation of various uncertainties in the tank vehicle stabilization system and the feedforward compensation. The proposed controller has excellent performance in the dynamic position command. It has strong adaptivity, which can adapt to the changes of the system working state and the perturbations of the external environment to maintain good control effect. Furthermore, it can effectively uphold stability and performance even in the presence of uncertainties.

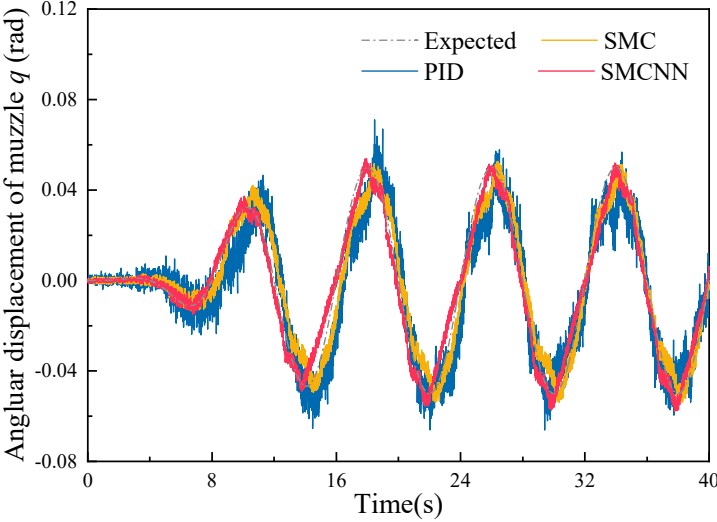

**Figure 25.** Dynamic-position angle tracking results.

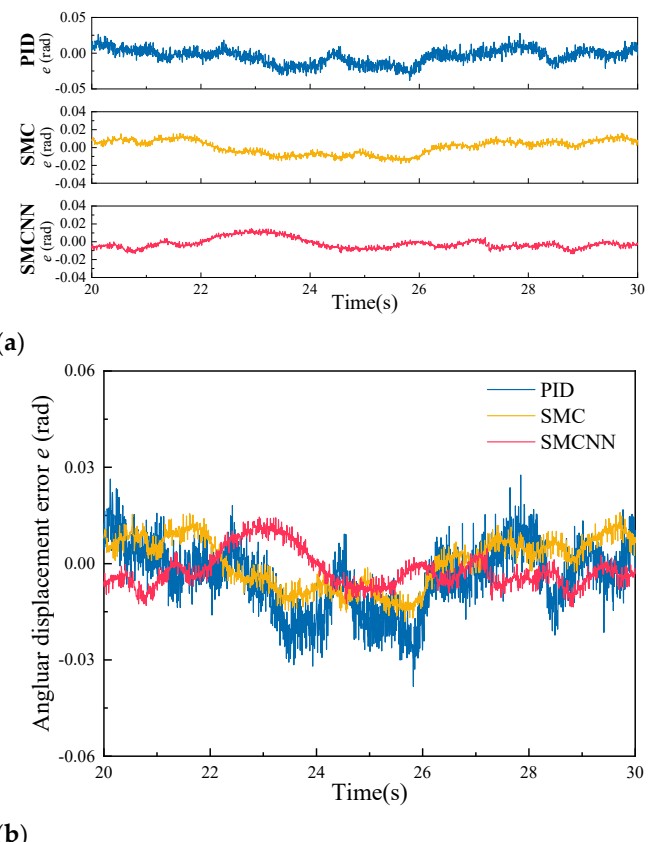

(**a**)

(**b**)

**Figure 26.** Dynamic-position angle tracking error: (**a**) errors of the three controllers; (**b**) comparison.

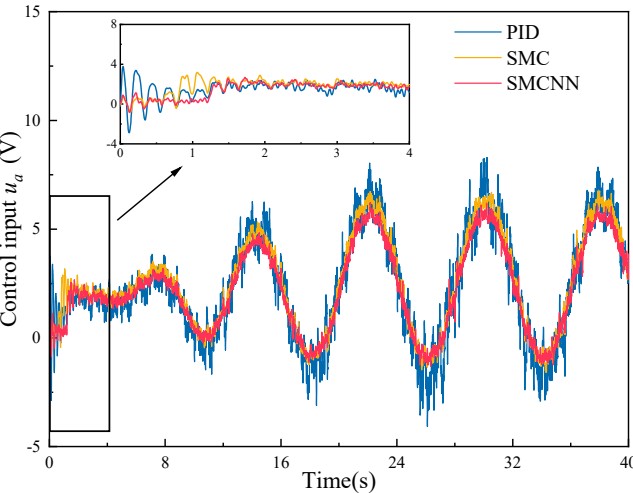

**Figure 27.** Comparison of control input for Case 2 in experimental verification.

In order to more intuitively present the control performance of several controllers, the literature [33] uses the three calculated values $M_q$, $\mu_q$, and $\sigma_q$ related to the angular tracking error as a measure of the quality of the controller, defined as

$$
\begin{aligned}
M_q &= \max\{|q_1|, |q_2|, \ldots, |q_n|\} \\
\mu_q &= \frac{1}{n} \sum_{i=1}^{n} |q_i| \\
\sigma_q &= \sqrt{\frac{1}{n} \sum_{i=1}^{n} (q_i - \bar{q})^2}
\end{aligned}
\tag{43}
$$

where $n$ is the count of samples and $\bar{q}$ is the average value of $q_1, q_2, \ldots, q_n$.

For the object of analysis in this paper, $\mu_q$ can be defined as characterizing the stable accuracy of the tank weapons. As a result, the calculation can obtain the control performance indexes of the three controllers at 20 km/h tank speed on E-grade pavement, as shown in Figure 28.

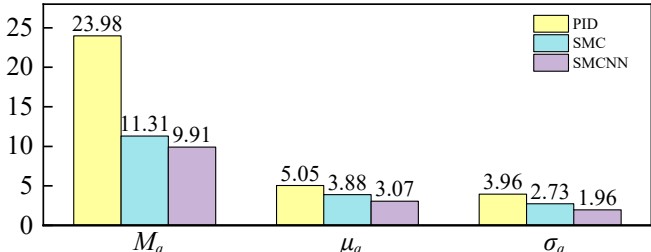

**Figure 28.** Performance comparison of three controllers.

In order to verify the robustness and adaptability of the controller designed in this paper, the pavement excitation is reconfigured, and the pavement excitation data of the tank system at the D-grade and F-grade at the driving speed of 20 km/h are obtained, respectively, and then the experiment is carried out again.

The experiment is conducted as described. Stability accuracy of the three controllers under the three pavement excitations can be calculated according to Equation (43). The experimental results are shown in Figure 29.

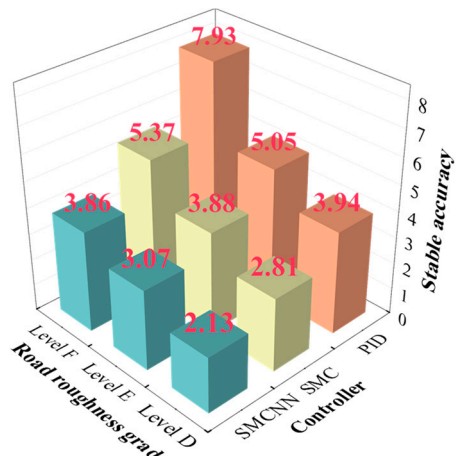

**Figure 29.** Controller performance under different pavement vibration excitations.

It is clear that the SMCNN controller proposed in this paper has better control performance than the other two controllers under the same working conditions. In addition, with the increase in external disturbance, the performance of SMC and PID controllers gradually becomes poorer, while SMCNN controller has higher robustness and stability, which are superior to the previous two controllers. It has strong engineering practicality and can be applied to different types of systems and working environments. At the same time, it can be automatically adjusted to maintain good control effect when external working conditions change. This again fully demonstrates the effectiveness of the nonlinear robust controller proposed in this paper that combines sliding mode control with neural networks for tank vehicle stabilization systems.

## 6. Conclusions

In this paper, a nonlinear robust control strategy based on neural networks is proposed for vehicle stabilization systems with uncertainty. The specific conclusions are as follows:

(1) The mechanism nonlinearity of the linear actuator transfer process is analyzed, and an accurate electromechanical coupled dynamics model for tank vehicle stabilization system that considers uncertainty is established.

(2) A sliding mode control strategy based on multi-layer neural network can effectively suppress the unmodeled dynamics and external disturbances in the tank vehicle stabilization system in high-mobility environments.

(3) The proposed control method naturally combines the advantages of sliding mode control and neural networks and proves the asymptotic tracking performance of the closed-loop system through strict stability analysis.

(4) The effectiveness of the proposed controller is verified by co-simulation and comparison experiments. Extensive experimental results show that the controller has excellent tracking performance and fast convergence.

**Author Contributions:** Conceptualization, Y.W.; methodology, Y.W., S.Y., G.Y. and X.W.; software, Y.W. and S.Y.; validation, Y.W., S.Y., G.Y. and X.W.; formal analysis, Y.W. and X.W.; investigation, Y.W. and X.W.; resources, Y.W.; data curation, Y.W. and S.Y.; writing—original draft preparation, Y.W. and S.Y.; writing—review and editing, G.Y. and Y.W.; supervision, X.W. and G.Y. All authors have read and agreed to the published version of the manuscript.

**Funding:** This work was supported in part by the Fundamental Research Funds for the Central Universities under Grant NS2022020, the China Postdoctoral Science Foundation under Grant 2020M671494, the Jiangsu Planned Projects for Postdoctoral Research Funds under Grant 2020Z179, and the National Natural Science Foundation of China under Grant 52175099.

**Data Availability Statement:** The authors confirm that the data supporting the findings of this study are available within this article.

**Acknowledgments:** My sincere and hearty thanks and appreciation go to my supervisor, Guolai Yang, whose suggestions and encouragement have given me much insight into these translation studies. It has been a great privilege and joy to study under his guidance and supervision. Furthermore, it is my honor to benefit from his personality and diligence, which I will treasure my entire life. My gratitude to him knows no bounds.

**Conflicts of Interest:** All authors declare that this research was conducted in the absence of any commercial or financial relationships that could be construed as potential conflicts of interest.

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
