# Peer review of "Nonlinear Robust Control of Vehicle Stabilization System with Uncertainty Based on Neural Network"

_electronics, doi:10.3390/electronics13101988_

Round 1

Reviewer 1 Report

Comments and Suggestions for Authors

Please see the attached review.

Comments on the Quality of English Language

Please see the review for some possible improvements.

Reviewer 2 Report

Comments and Suggestions for Authors

1. This paper presents the development of a dynamic model for a vehicle stabilization system, taking into account the non-linearity and uncertainties arising during operation.

2. The method proposed by the authors combines SMC and neural networks to estimate and compensate for system uncertainties. This compensation is necessary to improve the robustness and performance of the system.

3. To validate the effectiveness of the control strategy, the authors used a series of simulations and experimental studies.

Novelty of the paper:

1. Neural network integration is a relatively new approach for estimating the perturbations and dynamics of combat vehicle stabilization systems. This approach adds new information to the existing literature.

2. The use of adaptive neural networks for adjusting control parameters in real time brings some new elements. This approach can effectively cope with time-varying nonlinearities and uncertainties.

Week points:

1. The complexity of the proposed algorithms and the need to train neural networks may limit their applicability in real systems where computational resources are limited.

2. Although the paper validates the method on a specific model of a combat vehicle stabilization system, it remains unclear how well this approach can be generalized to other vehicle types or operating conditions.

3. The performance of the algorithm depends significantly on the quality and relevance of the data used to train the neural network model. This poses a challenge in real-world combat environments, where data can be noisy, contain perturbations or be incomplete.

Changes to be made to the work:

1. The terminology "electric cylinder" is not quite correct, the terminology "linear actuator" can be used instead.

2. More details are needed about the choice of PID controller parameters.

More tests and results on different values for the parameters Kp,Ki and Kd need to be presented.

Round 2

Reviewer 1 Report

Comments and Suggestions for Authors

The authors took into account all remarks and suggestions.

The most important changes are the simulations done for different parameter values and the simulations done for several frequencies of the reference signal.

Reviewer 2 Report

Comments and Suggestions for Authors

The authors of the paper have taken into account the recommendations made in the first phase of the review.

The paper may be published. Be careful with equations 22, some characters are missing.